# Effect of Microwave Radiation on Regeneration of a Granulated Micelle–Clay Complex after Adsorption of Bacteria

**A. Uğur Kaya [1], Selahaddin Güner [2], Marklen Ryskin [3], Azaria Stephano Lameck [3,4], Ana R. Benitez [3], Uri Shuali [3] and Shlomo Nir [3,*]**

[1] Department of Physics, Faculty of Arts & Sciences, Kocaeli University, 41380 Umuttepe-Kocaeli, Turkey; augurkaya@gmail.com

[2] Department of Chemistry, Faculty of Arts & Sciences, Kocaeli University, 41380 Umuttepe-Kocaeli, Turkey; seguner@kocaeli.edu.tr

[3] The R.H. Smith Faculty of Agriculture, Food and Environment, University of Jerusalem, Rehovot 76100, Israel; markryskin@gmail.com (M.R.); azarias.lameck@mail.huji.ac.il (A.S.L.); ana.benitez@mail.huji.ac.il (A.R.B.); ushuali@gmail.com (U.S.)

[4] Department of Natural Sciences, Mbeya University of Science and Technology, PO Box 131, Mbeya, Tanzania

[*] Correspondence: Shlomo.Nir@mail.huji.ac.il; Tel.: +972-8-9489172 or +972-54-8820412

**Abstract:** Granulated micelle–clay complexes including the organic cation octadecyltrimethylammonium (ODTMA) were shown to be efficient in removal of total bacteria count (TBC) from water. Microwave (MW) heating of granules to restore bacterial removal was investigated. Drying of granules by MW required 20-fold less energy than by conventional heating. When water content of granules approached 10%, or less, their heating period by MW had to be below 1 min, e.g., 30 s, and less, in order to avoid ignition and irreversible structural changes. Structural and thermal properties of MW heated samples were studied by FT-IR spectra and thermo gravimetric analyses (TGA). Inactivation of bacteria in water was more efficient by MW than by conventional oven, or by electric plate. For elimination of bacteria from water, MW heating was at least five-fold more efficient than by conventional heating. The results have established an adequate regeneration procedure by MW heating at durations depending on the remaining percentage of water associated with the granules. Tests of first and second regenerations by MW heating, and HCl washing of columns, were carried out. It was concluded that MW treatment may be chosen for optimal regeneration of the granulated micelle–clay complex as an efficient and low-cost procedure.

**Keywords:** bacterial removal; granulated micelle–clay; filter regeneration by microwave

## 1. Introduction

A general goal of drinking water treatment is to reduce health hazards due to pathogenic microorganisms in water using minimal concentrations of disinfectants. Chlorination at high doses of water, which includes organic molecules at concentrations of the order of mg/L, results in production of trihalomethanes (THMs) and haloacetic acids (HAAS), whereas reduced efficiency in eliminating some epidemic microorganisms occurred at low doses [1]. Disinfection by chloramines results in formation of nitrosamines [2]. Ozone is powerful in removal of microorganisms, but its application results in formation of nitrosamines [3] and cyanogen halides [4].

Filtration by means of a granulated micelle–clay complex has been proposed as one possible solution for reducing harmful disinfection byproducts (DBP). Two types of granulated complexes were employed. The micelle–clay complexes were synthesized by interacting micelles of an organic cation with a clay, which has a relatively large surface area. The two organic cations chosen were

octadecyltrimethylammonium (ODTMA), and benzyldimethylhexadecylammonium (BDMHDA), whose critical micelle concentrations were relatively small. They include large hydrophobic fractions and are positively charged to about half of the cation exchange capacity (CEC) of the clay, which is useful in the removal of microorganisms whose external surfaces are net negatively charged. Shtarker-Sasi et al. [5] employed micelle–clay complexes in powdered form, and demonstrated in batch and filtration experiments the removal from water of several types of Gram-negative and Gram-positive bacteria and *Cryptosporidium parvum*. Undabeytia et al. [6] demonstrated efficient removal of Escherichia coli (*E. Coli)* by a polymer-clay powdered complex. However, in a filter the powdered complex had to be mixed with an excess of a granulated material, such as sand, whereas the granulated complex can be included exclusively in filters, which enables upscaling. Recent studies demonstrated efficient removal of (i) pathogenic and indicator bacteria [7,8], and (ii) cyanobacteria [9], by filling filters with granulated micelle–clay complexes.

In removing pollutants from drinking water or wastewater, it is important to reduce the processing time and costs. Regeneration of the used filter after removal of bacteria was accomplished by passing dilute solutions of NaOCl or HCl, [5–8], or by heating in an oven for 2 h at 105 °C [7,8].

Microwaves are in the 300 MHz to 300 GHz frequency range of the electromagnetic spectrum. When microwaves interact with dielectric materials, the materials are heated as a result of the high frequency oscillations of the electric dipoles in the polar molecules. Due to penetration capability of microwaves, microwave (MW) heating is volumetric heating with advantages such as easy start-up and stopping, rapid heating, and non-contact heating. Bozkurt-Cekmer and Davidson [10] reviewed microbial inactivation by MW. A major aim of the current study was to explore the use of MW radiation for the regeneration of granulated micelle–clay complexes after being used in filtration of drinking water. This is the first study of the application of MW radiation to the regeneration and reuse of clay-composites in removal of bacteria from water by filtration. As will be shown, the MW procedure proved to be optimal in terms of removal of bacteria and saving of energy and time. The regeneration, which enables a low cost reuse of the purification complex, has an environmental advantage.

## 2. Materials and Methods

### 2.1. Materials

Bentonite was purchased from Tolsa–Steetley, Guisborough, UK. The organic cation ODTMA was purchased as a bromide salt from Sigma-Aldrich (Sigma Chemical Co., St. Louis, MO, USA). The organic cation BDMHDA was purchased as a chloride salt from Fluka Chemie, Buchs, Switzerland.

Non-woven polypropylene geo textile filter was from Markham Culverts Ltd, Lae, Papua-New Guinea.

### 2.2. Granulation Procedure

Granulated complexes of ODTMA– and BDMHDA–clay of diameter sizes between 0.3 and 2 mm were prepared as described [11,12]. The granules included 3% of water (see Supplementary Material).

### 2.3. TBC

Total bacteria counts (TBCs) were performed by a certified laboratory, Aminolab, Ness Ziona, 70400, Israel, in accordance with the Standard Methods [13].

### 2.4. Filtration Experiments of TBC

Water was taken continuously from a tap into a container, which was overflown to avoid reproduction of bacteria. The filters were fed from the container by a peristaltic pump. The filters used (1.6 cm diameter × 20 cm length) were filled exclusively with 32 g of a granulated complex. Non-woven polypropylene geotextile filters (Markham Culverts Ltd., Lae, Papua, New Guinea) were placed on both ends of the column. The flow rate was 6 mL/min, i.e., a flow velocity of 1.8 m/h. Prior

to each experimental run, water was added to the columns at a slow rate in an upward direction in order to eliminate air pockets and channeling. Each experiment was conducted in triplicate at least.

## 2.5. Regeneration

Regeneration of filters (1.6 cm × 20 cm) containing 32 g of micelle–clay complexes after filtration of TBC was carried out by passing 250 mL of a solution of 0.05 M HCl followed by washing with 500 mL of tap water, or/and by heating. Heating of filter was applied in an oven or a microwave oven; mainly two procedures were applied: (i) heating in an oven at 120 °C for 2 h, or (ii) heating in a MW oven for a variable number of minutes, or less, depending on residual humidity, as described in Section 3. For the columns used in the current study, the wet granules from one column were removed into a small MW compatible bowl and weighed. The bowl was introduced into the MW oven for 30 to 60 s operation at 700 W. Then the granules were mixed and weighed again. In the second round the period of MW operation was reduced, e.g., to 30 s. Each consecutive round involved shorter periods of heating, until the original weight of 32 g was approached. The same procedure was applied to all the columns. It was also possible to use in one round granules from more columns, and correspondingly enhance (not linearly) heating times.

## 2.6. Drying in Ovens

Granulated micelle–clay complexes were subjected to MW irradiation to determine optimal drying times of complexes. The domestic MW oven, model P70B17-A3 (Galanz, Foshan, Guangdong, China) was characterized by a maximal power of 850 W and a frequency of 2.45 GHz. The sample mass was measured after each heating–drying period, and the onset of degradation of the granules was determined. For comparison, an ordinary oven used for heating and drying was HEM123, 230 V/50 Hz with maximal power of 1800 W, which was typically operated at 800 W.

In one type of experiment, 33 g (or more) of granulated micelle–clay complexes, which included 32 g of complex with 3% humidity, plus 1 g of added water (in this case) were subjected to MW irradiation. Temperature measurements employed a thermometer and thermocouple. Sample preparation included drying of samples to a constant mass determined by observing that the mass did not change during the last three measurements. After heating, the sample was cooled at room temperature and weighed.

## 2.7. Characterization Techniques

Infrared spectra (FT-IR) of the samples were recorded on a Perkin Elmer Spectrum Two, FT-IR spectrometer (Perkin elmer Spectrum Two, Infrared Spectrometer, Llantrisant, Pontyclun, United Kingdom). The pellet was prepared by pressing finely ground powder of 4 mg of sample and 400 mg of dried KBr; 200 mg of sample–KBr powder was pressed under a hydraulic press of 5 tons to produce KBr pellet. The spectra were acquired in the range of 4000–400 $cm^{-1}$ with a scan rate of 1 $cm^{-1}$. A houseware MW oven (Samsung Electronics, Selangor Darul Ehsan, Malaysia) was operated at 850 W power and 2.45 GHz frequency (Turkey). The MW heating time was 450 s.

Thermogravimetric analyses (TGA) were carried out on a Perkin Elmer Pyris thermal analyzer, at temperatures ranging from 30 to 820 °C, at a heating rate of 10 °C/min and under a nitrogen flow (20.0 mL/min).

## 3. Results

### 3.1. Drying of Granules

A sample of 5 grams of ODTMA– or BDMHDA–clay granules was subjected to the MW irradiation for 90 s, 180 s, 270 s, 360 s, and 450 s. After each duration, sample mass was measured. The sample was then immediately placed in the MW oven for another duration. After 360 s of MW irradiation of ODTMA–clay sample, mass loss was about 2.3% (Figure 1). Mass losses of BDMHDA–clay were essentially similar, and will be discussed later. It has been observed [14,15] that MW heating caused

larger mass losses in several clays and organo-clays than conventional heating. MW heating makes it possible to easily remove weakly bound water molecules. These water molecules are weakly retained between the surfactant ion layers as previously explained [16]. In addition, the moisture in the surface is easily removed by MW. In the current study, after about 6 min of MW heating the complexes started to deteriorate.

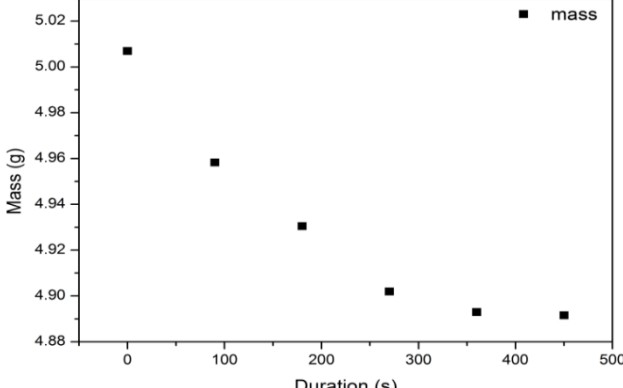

**Figure 1.** Mass loss of octadecyltrimethylammonium (ODTMA)–clay sample under microwave (MW) irradiation.

Results of drying of granules based on ODTMA and BDMHDA in a conventional oven at 60 °C yielded after 30 min removal of 0.2 ± 0.1 g water from 32 g granules (including 3% humidity) to which 1 g of water had been added. A larger removed amount of 0.9 ± 0.1 g water was observed by MW heating for 90 s of ODTMA–clay granules (whose temperature was 145 °C). A smaller loss of 0.6 ± 0.1 g was observed for BDMHDA–clay granules (at 135 °C).

The results (Table 1) indicate that in comparison to an ordinary oven, removal of water was achieved by the MW oven within about a 20-fold shorter time. It is noted that the ordinary oven was operated at a maximum of 800 W, whereas the MW was operated at 700 W.

**Table 1.** Results of heating of ODTMA–clay wet granules (65% solid) in an ordinary oven (110 °C) and in MW oven (80–100 °C).

| Conventional Oven [a] | | | | MW Oven | | | |
|---|---|---|---|---|---|---|---|
| Time (h) | Granules (g) | | | Time (min) | Granules (g) | | |
| 0.0 | 61.5 | 92.3 | 123.1 | 0.0 | 61.5 | 92.3 | 123.1 |
| 0.5 | 57.1 [b] | 84.7 | 111.8 | 5.0 | 41.2 [b] | 69.7 | 100.4 |
| 1.0 | 52.2 | 79.5 | 110.2 | 7.0 | 39.3 | 64.3 | 87.3 |
| 1.5 | 45.7 | 71.0 | 102.1 | 8.0 | - | 62.3 | 80.6 |
| 2.0 | 40.0 | 64.1 | 88.1 | 9.0 | - | 60.0 | 79.8 |
| 2.5 | - | 59.8 | 84.7 | - | - | - | - |
| 3.0 | - | - | 80.2 | - | - | - | - |

[a] The ordinary oven was operated at 800 W, whereas the MW was operated at 700 W. The standard error of each mass was less than 1 g. [b] The values in columns 2–4 and 6–8 in the line starting with 0.0 are initial masses (g) of the granules before heating.

The results in Table 2 demonstrate that MW heating removed water more efficiently from ODTMA–clay granules than from BDMHDA–clay granules whose critical micelle concentrations are 0.3 and 0.6 mM, respectively. This result is consistent with the outcome that the magnitude of free energy of interaction of ODTMA cations with the clay surface exceeds that of BDMHDA cations. This was seen by the smaller extent of release of ODTMA cations from the complex than that of BDMHDA cations in suspension [17] and during filtration [8]. It can be rationalized that the presence

of more water molecules reduces the magnitude of the interaction between the organic cations and the surface sites.

**Table 2.** Drying in MW oven: Granules included ODTMABr or BDMHDACl [a].

| ODTMA Br | | | | BDMHDA Cl | | | |
|---|---|---|---|---|---|---|---|
| Time (min) | Mass g | Time (min) | Mass g | Time (min) | Mass g | Time (min) | Mass g |
| 0 | 78.5 | 11 | 47.6 | 0 | 78.5 | 11 | 50.1 |
| 2 | 74.3 | 12 | 45.0 | 2 | 74.5 | 12 | 47.1 |
| 4 | 69.5 | 13 | 42.5 | 4 | 70.6 | 14 | 44.7 |
| 6 | 63.7 | 14 | 41.1 | 6 | 64.6 | 16 | 42.9 |
| 8 | 57.5 | 15 | 40.7 | 8 | 58.2 | 17 | 41.5 |
| 10 | 50.7 | 16 | 40.2 | 10 | 52.7 | 18 | 41.0 |

[a] Conditions as in Table 1.

In the following experiments the MW oven was operated at 700 W and drying of granules with a water content of 5% to 40% was studied. The results (Table 3) demonstrate that at a humidity of less than 30%, a sharp increase of temperature and the ignition point of the material was reached. By reducing the heating time from 2 min to 1 min (or less), the ignition could be avoided.

**Table 3.** Heating of granules (ODTMA–clay) which included different water percentages in a MW oven (700 W) for several durations: removal of water and elevation of temperature of granules [a].

| Humidity [%] | Heating Time [min] | Sample Mass [g] | Temp [°C] | Remark |
|---|---|---|---|---|
| 5 | 0 | 27.4 | 26 | |
| 5 | 2 | 24.3 | >100 | Material was ignited |
| 20 | 0 | 32.5 | 26 | |
| 20 | 2 | 26.6 | 88 | Material was ignited |
| 40 | 0 | 43.3 | 26 | |
| 40 | 2 | 34.3 | 85 | |
| 40 | 3 | 29.7 | 88 | |
| 40 | 4 | 26.6 | 98 | |

[a] The standard error of each mass was less than 0.3 g.

### 3.2. Changes in Granule Morphology upon Heating

The shape of the granules before and after heating for 4 min in a MW oven and for 90 min in a conventional oven at 120 °C is shown in Figure 2. It is demonstrated that 4 min of heating of both spherical (which are more robust) and non-spherical granules in a MW oven caused a certain degree of destruction of the structure. In this context, it is noted that in the course of heating of granules by MW radiation, cases were encountered (e.g., Table 3) in which prolonged heating caused ignition of the granules.

### 3.3. Infrared Study

Infrared spectra of ODTMA–clay and BDMHDA–clay samples are shown in Figures 3 and 4, respectively. The peak observed at 3628 cm$^{-1}$ is related to structural water (Figure 3a). No change in this peak was observed (Figure 3b). Thus, after MW heating the structural water between the clay layers was not affected and the layers did not collapse. The weak peak observed at 3411 cm$^{-1}$ is related to the absorbed water (stretching vibrations) (Figure 3a). This highly weakened peak represents the hydrophobic structure of the organo-clay resulting from the interaction of the surfactant molecules with the clay. This interaction is described as the interlocking of the siloxane layer of the clay and surfactant methyl group [18]. The spectral profile region changes significantly with the concentration of surfactant molecules in the 3000 to 3550 cm$^{-1}$ range [19]. The intensity of this band decreases with surfactant concentration and is attributed to water hydrating the cation in the clay interlayer [19]. The very low peak intensity observed at 3411 cm$^{-1}$ is due to the high concentration of ODTMA in the organo-clay. In addition, the wavenumber of antisymmetric CH$_2$ stretching mode is sensitive to the

conformational change of amines within the clay interlayer [19]. This peak was observed at 2917 cm$^{-1}$ for the ODTMA–clay sample, whereas it was observed at 2921 cm$^{-1}$ for the BDMHDA–clay sample (Figure 4). The different adsorbed water behaviors in both organo-clays (ODTMA and BDMHDA) are attributed to the difference in concentrations of organic cations, or surface interaction mechanisms. This peak disappeared after MW heating. Therefore, a small amount of adsorbed water in the organo-clay was removed from the samples by MW heating. The peaks observed at 2917 cm$^{-1}$ (asymmetric) and 2849 cm$^{-1}$ (symmetric) are related to the C-H stretching vibrations. In these peaks small changes were observed with MW heating. The peak observed at 1642 cm$^{-1}$ represents the bending vibration of adsorbed water. This peak may originate from residual water molecules between the ODTMA ion layers as mentioned previously [16]. This peak was weakened upon MW heating (Figure 3b). The Sharp peak at 1012 cm$^{-1}$ is due to Si-O stretching. Small changes in this peak were observed. It was indicated that the CH$_2$-stretching vibration of amine chains was very sensitive to the conformational ordering of the chains [20]. Small variations in both C-H peaks (1917 cm$^{-1}$ and 2848 cm$^{-1}$) and Si-O (1012 cm$^{-1}$) peaks are due to the alteration of the conformation of the surfactant chains as water molecules are affected by the MW. Therefore, the shifts in the peaks are apparent in the MW irradiated sample.

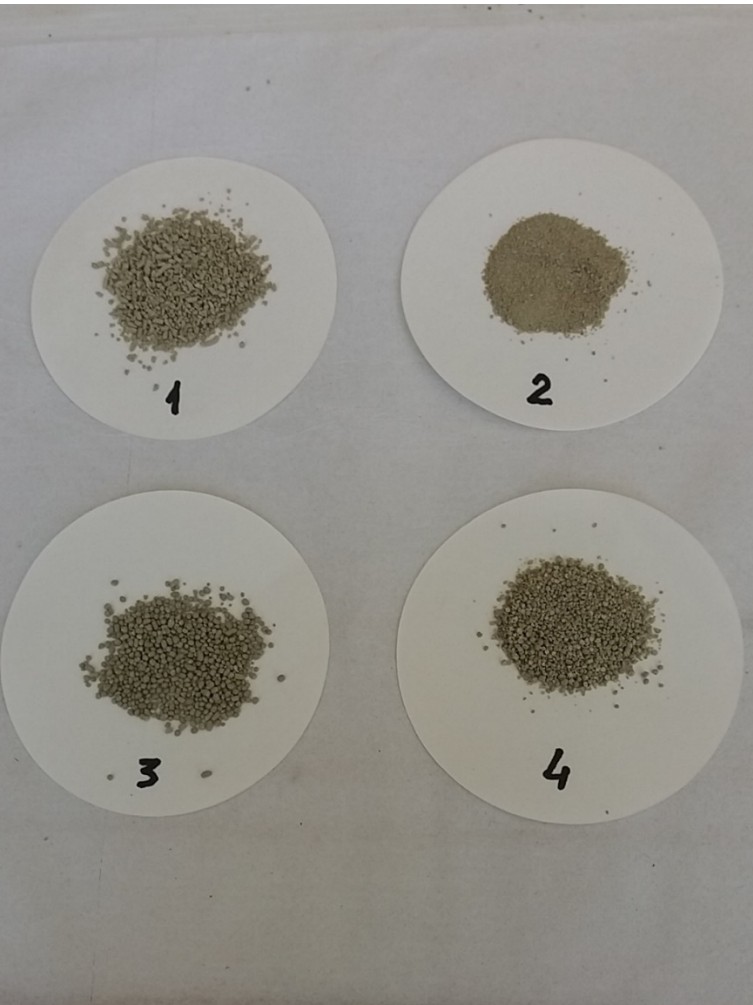

**Figure 2.** Effect of heating on morphology of micelle–clay granules. 1. Non-spherical granules—drying for 90 min in an ordinary oven. 2. Non-spherical granules—drying for 4 min in a MW oven. 3. Spherical granules—drying for 90 min in an ordinary oven. 4. Spherical granules—drying for 4 min in a MW oven.

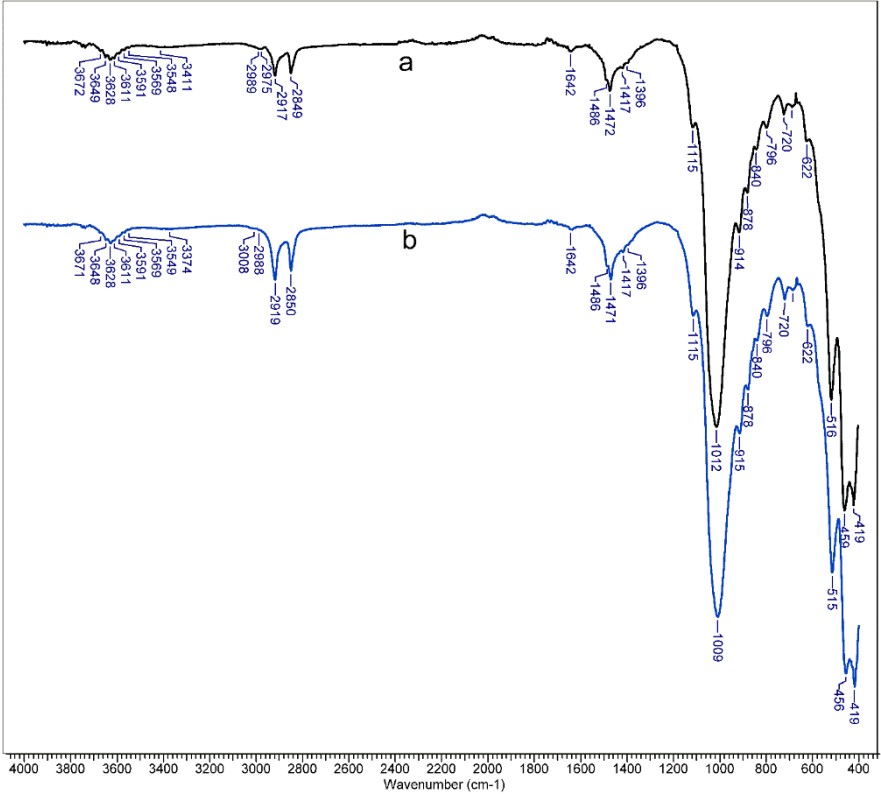

**Figure 3.** FT-IR spectra of ODTMA–clay sample under: (**a**) MW free; (**b**) MW heated.

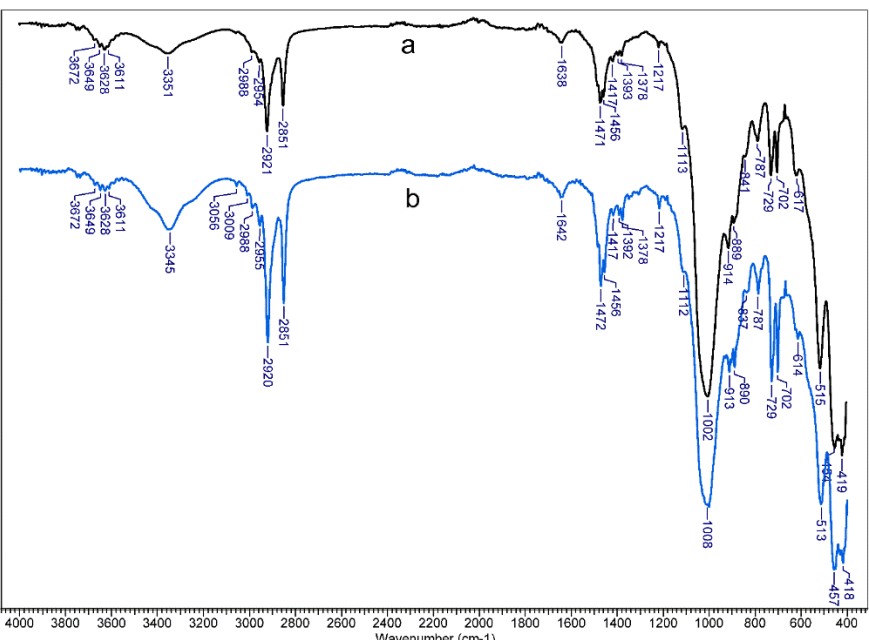

**Figure 4.** FT-IR spectra of BDMHDA–clay sample under: (**a**) MW free; (**b**) MW heated.

Complementary information can be obtained from a study by FT-IR on the adsorption of ODTMA as monomers and micelles by the clay-mineral Na-montmorillonite [21].

As in the case of ODTMA–clay, no change was observed in the structural water peak observed at 3628 cm$^{-1}$ in the BDMHDA–clay sample (Figure 4). The broad peak observed at 3351 cm$^{-1}$ arises from H-bonded water molecules. This broad band (H bound water), which is generally seen in a high concentration of surfactant, shifts to a low wave number with increasing concentration [22]. This peak

shifted to 3345 cm$^{-1}$ after microwave heating. The decrease in band intensity is an indication of low levels of water molecules, which indicates low water content [22]. In this study, the increase in the intensity of this peak in the MW heated sample indicates the change of H-bonded water (Figure 4). This may be due to the increase in pore sites and diameters with the removal of existing water by MW heating. Thus, a greater amount of water settles on the surface with increasing pore diameters. The peaks observed at 2921 cm$^{-1}$ and 2951 cm$^{-1}$ are related to the antisymmetric and symmetric CH$_2$ stretching modes of amine, respectively (Figure 4). The narrow peaks and the increase in the intensity represent the increased density of the amine chain and the more regular amine chains [22]. The intensity of these peaks increased and the band width narrowed. The increase in peak intensity is high in the MW heated sample. The peak observed at 1638 cm$^{-1}$ is related to adsorbed water. This peak shifted to a higher wave number by MW heating, (Figure 4). The shift of adsorbed water peak to higher wave numbers represents high surfactant concentration and thus lower adsorbed water content [22].

### 3.4. Thermogravimetric Analysis

Figure 5 shows thermogravimetric analysis of ODTMA–clay samples. The first decomposition step is related to the mass loss of adsorbed water and gaseous species (<150 °C). The lowest mass loss was observed in MW heated samples (0.31%, at 100 °C) with respect to other samples. Mass loss in the sample not subjected to MW heating (MW free) was 1.53 percent. The weakly bound water molecules in the pores between the interlayer ODTMA ions are expected to be the first target of the energy that the MW will use for the dipole orientation. As a result of the large magnitude of the energy of dipole orientation, which is the basis of dielectric heating, gaseous species and weakly bound water molecules were apparently detached. In the MW heated sample, the observation of low mass loss indicates that weakly bound water between the organic molecule and the clay surface did not form pores again after heating. As a result, MW heating increased the level of hydrophilicity of organo-clay.

Onset temperatures were shifted to the lower values by MW heating. A low temperature endothermic peak was observed at 65 °C for the MW free sample. The endothermic peak observed in natural bentonite at around 100 °C has been reported to shift to 70 °C in organic bentonite [15]. It was noted that MW heating is an internal process [23].

The second region in TGA curves (150–550 °C) is associated with the organic substances. In this region, the surfactant begins to decompose [15]. The initial decomposition temperature of ODTMA is 272 °C for a MW free sample. In MW heated samples, this temperature decreased to 264 °C. In the process of removing the impurities in the organo-clay used in water filtration and inactivation of bacteria, MW heating would be sufficient. In the second part of the dehydration stage (120–200 °C), elimination of adsorbed water still remains effective up to 200 °C [18]. The secondary thermal degradation step (420 °C) could be attributed to the desorption of surface hydration water, the loss of structural hydroxyl groups, and a crystal phase change [14,24]. Although this temperature step did not change in the case of the MW free sample (420 °C), the MW heated sample showed slight variations (417 °C). Thus, a small portion of the mass loss in this region is due to the removal of water and the majority of the mass loss is due to the decomposition of the surfactant. In this region, MW heating processes resulted in high mass loss compared to the ODTMA clay sample (MW free sample).

The third region, which is called dehydroxylation of clay (structural water), is between 550 and 700 °C [15,18,25,26]. In the MW heated sample, the mass loss did not change significantly compared to the ODTMA clay sample (MW free sample).

The corresponding results for BDMHDA–clay are similar to those of Figure 5a,b, particularly in the range of temperatures up to 150 °C. For the sake of completeness, these results are presented in Figure S1a,b.

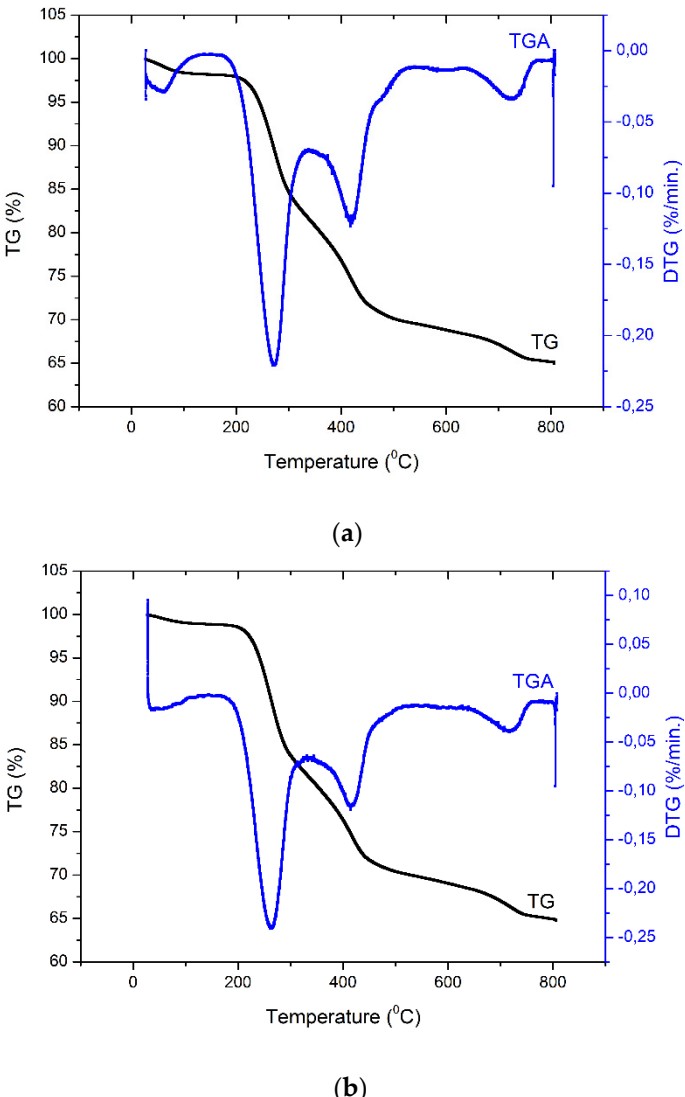

**Figure 5.** (**a**) Thermogravimetric analysis of ODTMA–clay samples; no MW; (**b**) thermogravimetric analysis of ODTMA–clay samples with MW treatment.

### 3.5. Removal of Bacteria from Water Filtered by Saturated Granules

Three samples were prepared, each with water filtered through a saturated micelle–clay column. Within experimental error, initially all samples were expected to include the same numbers of bacteria. Sample M1 was not treated. The tests indicated that it included 5900 TBC per mL. Sample M2 underwent heating to a temperature of 90–98 °C for 22 min by placing it in a vessel on an electrical plate operated at 800 W. The analysis gave 2 TBC per mL. Sample M3 underwent heating in a MW oven operated at 140 W for 26 min. The test indicated <1 TBC per mL. It is remarkable that complete elimination of the bacteria from water was energetically at least five-fold more efficient than by conventional means.

### 3.6. Filtration of Tap Water before and after Regenerations

As described in Section 2.4, the filter columns included 32 g of granulated ODTMA–clay. The focus in these experiments was to test the possible efficacy of microwave heating on the regeneration of the used granules, which were fully or partially saturated after filtration of tap water. Tables 4–6 demonstrate the numbers of emerging TBC per mL after a given number of days or volumes of filtration

by fresh granules, and after first and second regenerations, respectively. The average volume filtered continuously per day was 8.64 L. These tables represent results from several sets of such experiments.

**Table 4.** Removal of total bacteria count by fresh micelle–clay granules: number of emerging bacteria (CFU per mL) after filtration of given volumes [α].

| Column ID | 37.5 (L) | 64 (L) | 78 (L) |
|---|---|---|---|
| TAP | 1200 [a] | 1400 [a] | 40 [b] |
| Container | 1100 [a] | 3400 [a] | 70 [b] |
| M1 | | 9 | 12 |
| M2 | | 1 | 28 |
| M3 | | 835 [a] | 925 [a] |
| M4 | | <1 | 1 |
| H1 | | 5 | 5 |
| H2 | | 1 | 363 [a] |
| H3 | | - | 56 [b] |
| H4 | | 2700 * | 23,000 * |
| Av of H samples | 1 | 902 [a] | 5856 * |
| Av of M samples | 350 [a] | 211 [a] | 242 [a] |

[α] Experiment was in September, 2019. TAP and Container refer to continuous tap and container water inlets, respectively. On the first sampling (37.5 L) all samples denoted by H were mixed together and only the mixture was sampled; similarly, the same was done for samples denoted by M. The average values were also calculated for the samples taken after 64 and 78 L. The average flow rate was 6 mL/min for each column. The estimated relative errors in the number of bacteria in cases denoted by *, [a,b] are 50% or above, 30%, and 20%, respectively; in other cases, they are less than 10%.

**Table 5.** Comparison of removal efficiency of total bacteria count from tap water by ODTMA–bentonite granules after the first regeneration of the complex by passing dilute HCl solution (H) or by MW heating: number of emerging bacteria after passage of given volumes *.

| Source of Sample | 9 L | 43 L | 76 L |
|---|---|---|---|
| TAP | 40 [d] | 70 [d] | 150 [d] |
| Container | 40 [d] | 60 [d] | 160 [d] |
| M1 | 10 | 22 | 1500 [c] |
| M2 | 1 | 7 | 3 |
| M3 | 880 [c] | 30 [d] | 440 [c] |
| M4 | 1 | 30 [d] | 34,000 [b] |
| H1 | 8 | 10 | 160,000 [a] |
| H2 | 17 | 190 [c] | 250,000 [a] |
| H3 | <1 | <1 | 2600 [c] |

* The experiment began on September 25 and was completed by October 7, 2019. The regeneration was applied to the seven columns indicated in Table 4: by MW radiation (M1 to M4), or by passing a HCl solution (0.05 M) through columns H1 to H3. The estimated relative errors in the number of bacteria in cases [a,b,c,d], are 60%, 50%, 30%, and 20%, respectively; in the other cases, they are less than 10%.

The permitted use of the water for drinking was set at 1000 TBC per mL. The total volume that passed through each column filter was 78 L, which corresponds to a capacity of at least 2.4 liters per gram, since the columns were not saturated yet. This value is larger than 2 L/g in Kalfa et al. [8]. For quite a few experiments the capacity was larger than in Table 4 during spring and winter months. The capacity in March 2019 was 143 L, corresponding to 4.5 L/g (Table S1). The improved results in the current experiments are explained by prevention of contaminated water in the container and disinfection of the container and pipes before the beginning of the experiments. The results in Table 4 indicate that in the first days of the experiment, the tap water included a relatively large number of TBC, but the emerging water had significantly smaller values.

**First regeneration**. The average flow rate (Table 5) was 5 mL/min for each column filter. The total volume passed through each column filter was 76 L. For the first 43 L all the columns exhibited filtered water with permitted numbers of TBC. After filtration of 76 L, two of the four columns treated by MW

heating yielded permitted values, whereas all the filters treated by HCl were inadequate. After the passage of 76 L the treatment by HCl yielded, on average, an order of magnitude larger emerging numbers of TBC than after MW heating. For the MW treated columns the average permitted volume after first regeneration amounted to (43 + 33/2) L = 59.5 L and the capacity corresponded to 1.9 L/g. This value is larger than the corresponding value reported by Kalfa et al. [8].

**Table 6.** Comparison of removal efficiency of TBC by ODTMA granules after the second regeneration (CFU per mL): number of emerging bacteria after passage of given volumes *.

| Source of sample | 9 L | 31 L | 44 L |
|---|---|---|---|
| TAP | 5 | 690 [c] | 50 [d] |
| Container | 7 | 200 [c] | 140 [d] |
| M1-MW | <1 | 1100 [c] | 250 [c] |
| M2-HCl | <1 | 23 | 71,000 [b] |
| M3-MW | 13 | 980 [c] | 180 [c] |
| M4-HCl | 11,000 [b] | 260 [c] | 550,000 [a] |
| H1-MW | 8 | 11,000 [b] | 8,200 [b] |
| H2-HCl | <1 | 550,000 [a] | 250,000 [a] |
| H3-MW | 1 | 260[c] | 60 [d] |

* The experiment was completed by October 15, 2019. Columns 2–4 in the table indicate the filtered volumes. The notation describes the history of treatment, e.g., M2-HCl indicates that this sample was obtained from a column which underwent a first regeneration by MW heating and a second regeneration by washing with a dilute HCl solution. The estimated relative errors in the number of bacteria in cases [a,b,c,d], are 80%, 50%, 30%, and 20%, respectively; in other cases, they are less than 10%.

It may be added that Table S2 presents results of regeneration by MW of six columns of which two were saturated; then filtration was stopped for 20 days and resumed for one additional day. Samples from three columns showed emerging bacteria between 1300 and 67,000 per mL. Two days after regeneration, filtration through these columns yielded emerging bacteria in the range of 2 to 96 (average 29) per mL.

The average flow rate after the second regeneration (Table 6) was 5 mL/min for each column filter. During the first and last days of filtration, the flow rate was 6 mL/min. The total volume that passed through each column filter was 44 L. It is remarkable that the second regeneration enabled a reduction of the number of emerging TBC after the first day of filtration from 160,000/ mL (H1-MW) and 250,000/mL (H2-HCl) at the end of first regeneration (Table 5) to 8/mL and <1/mL, respectively. The analysis of the last sample after second regeneration (44 L) indicated that only MW treatment yielded permitted drinking water. The contribution of the second regeneration to the filter capacity amounted to 22.7 L, or 0.7 L/g.

A third regeneration was carried out only by MW heating of granules from four columns (Table S3). One day after the third regeneration all samples included small numbers of emerging TBC (average 32 per mL), whereas after the second day, three samples yielded acceptable values. It appears that low-cost regeneration by MW heating could increase the capacity of the filter at least two-fold.

## 4. Discussion

The issue of the mechanisms of MW inactivation of microorganisms and contribution of "non-thermal effects" was widely discussed and reviewed by Bozkurt-Cekmer and Davidson [10]. The application of MW at a sub-lethal temperature of 40 °C on *E. coli* for a duration of 1 min resulted in disruption of the permeability barrier, i.e., cell leakage and entry of large molecules, such as labeled dextran, into the cell were observed [27]. They noted that 10 min after terminating the MW radiation, the cell morphology reverted to that before the treatment, i.e., the MW treatment was not biocidal. It can be stated, however, that prolonged disruption of the permeability barrier at any temperature would lead to cell death. In the current work (Section 3.5) about five-fold less energy was required for

a biocidal or biostatic effect for TBC present in hot water (below boiling point) under MW operation than heating by an electric plate.

An evaluation of the maximal number of bacteria that could be retained by 1 g of the complex [5,8,9] indicates that the available area of the complex is several orders of magnitude larger than the total area of the external surfaces of bacteria. Hence, the limit on the number of bacteria which can be adsorbed is dictated by the balance between adsorption and desorption processes. For filtration of a single species of bacteria, a model of kinetics of filtration, which considered convection, adsorption, and desorption could simulate and yield predictions for the kinetics of bacteria filtration [5,6,8,9]. In the initial stages of filtration in the current experiments (on a time scale of days in the case of the filters used) only a few bacteria were found to be emerging through the filter, but eventually the pores in the filter included free bacteria, which could reproduce, and the numbers of bacteria emerging through the filter kept increasing with the volume passed and with time. At this stage the regeneration has to inactivate the bacteria residing in the water-filled pores, which in the current filters constitute about 50% of the volume of the active layers.

Regeneration can increase the shelf life and capacity of the filter and reduce the frequency of its replacements. In planning the current study, cost evaluations favored an initial step of washing the used filter by 250 mL of a dilute HCl solution (0.05 M) followed by 500 mL of tap water, rather than washing by Na-hypochlorite solution [5,8].

The present study was largely designed to test a hypothesis that the advantage of MW over ordinary oven heating in the regeneration of granulated micelle–clay complexes is due to a more efficient biostatic/biocidal effect on bacteria. During a search for an optimal regeneration that employs MW, a factor of prime importance was the finding that overheating of the complex, sometimes just for one minute, resulted in irreversible destruction of the gross granular structure, as well as the nanostructure of the complex. A breakthrough in establishing an optimal regeneration procedure based on MW heating emerged, by realizing that the periods of applied MW heating can be calibrated by the relative amount of water remaining associated with the complex, as demonstrated in Tables 1 and 3. The results in Table 1 demonstrate that heating by MW oven consumed 20-fold less energy than heating by an ordinary oven, and the corresponding required times were five-fold shorter, when considering the extra time used for a manual determination of the masses of samples before and after their drying, and mixing of the granulated complex. The results of first two regenerations demonstrate that the use of MW in regeneration is an efficient procedure. We have encouraging results that a third regeneration may also be considered. For two regenerations, the times used for MW heating were up to 7.2 min, which amounts to invested MW energy of less than 0.1 KWH. This corresponds to a cost of less than 3% of the granulated complex. Furthermore, environmental considerations should encourage the concept of regeneration.

## 5. Conclusions

The capacity of granulated micelle (ODTMA)–clay (bentonite) to remove total bacteria count from drinking water could be increased by an efficient regeneration procedure, such as MW heating. Thermal analysis and FT-IR results indicated that at temperatures applied in the current study (<150 °C) no deterioration of the structure of the complex occurred. However, a longer MW operation period, sometimes by just one minute, could result in clogging of the filter and irreversible destruction of the structure of the complex. An optimal regeneration procedure based on MW heating emerged by realizing that the periods of applied MW heating should be adjusted to the fraction of remaining water associated with the complex. In comparison with an ordinary oven, regeneration of used micelle–clay complexes after removal of TBC from drinking water by MW heating needed 20- and five-fold less energy and time, respectively. The regeneration by MW heating is an optimal low-cost procedure.

## 6. Patents

Nir, S., Ryskin, M., 2019. Method of production of granulated micelle–clay complexes: application for removal of organic, inorganic anionic pollutants and microorganisms from contaminated water. Patent 10384959 USA, 08/20/2019.

**Supplementary Materials:** The following are available online at http://www.mdpi.com/2076-3417/10/7/2530/s1. Figure S1. (a) Thermogravimetric analysis of BDMHDA–c-Clay samples, no MW; (b) Thermogravimetric analysis of BDMHDA–c-Clay samples with MW treatment; Table S1. Removal of total bacteria count by fresh micelle-clay granulated complex by filtration (CFU per mL) *; Table S2. Removal efficiency of total bacteria count by ODTMA–c-clay fresh granules: filtration of 8.6 L after stopping for 20 days followed by filtration after MW regeneration (CFU per mL); Table S3. Removal efficiency of TBC by ODTMA- granules after third regeneration (CFU per mL): number of emerging bacteria after filtration of given volumes *.

**Author Contributions:** Conceptualization, U.S., S.G., M.R., A.R.B., A.U.K., and S.N.; methodology, U.S, S.G., M.R., A.S.L., A.R.B., A.U.K., and S.N.; writing, U.S., S.G., M.R., A.R.B., A.U.K., S.N.; funding acquisition, S.N. All authors have read and agreed to the published version of the manuscript.

**Funding:** This research was partially supported by the Ministry of Science & Technology, Israel & The Ministry of Science and Technology of the People's Republic of China (grant No. 3-15707).

**Conflicts of Interest:** The authors declare no conflict of interest.

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
