# Peer review of "Effect of Microwave Radiation on Regeneration of a Granulated Micelle–Clay Complex after Adsorption of Bacteria"

_applsci, doi:10.3390/app10072530_

Round 1
Reviewer 1 Report
The paper could surely be of interest for the journal readers, but in my opinion it suffers from two main drawbacks:
One the one hand it is not clear the relation of this study with other studies on the same subject. Which is its innovative character? In my opinion the state of the art for what concerns the disinfection through clay complexes is not adequately described
On the other hand, methods are not adequately described. The results contained in most of the tables are of very difficult reading and interpretation.
Punctual remarks:
Line 45 – The authors mentioned two types of granulated complexes, but they did not explain what these types are. In the same line it is not clear to what “produced” refers.
For which reason the regeneration by using MW could be preferred to other, more traditional, methods? Not clear if it is a completely new method or if it was tested by other authors.
Line 76 – Please refer to Supplementary Materials for the description of the preparation of the granules.
Line 128 – Where are shown the results described from line 128 to line 132?
Table 1 – not clear the sense of the results reported in the Table. Which is the meaning of the figures reported in the three columns below “granules”?
Table 2 – a diagram should probably be a better representation of these data. In my opinion the difference between the two types of granules are not so evident. Did you perform out some replicates? Did you carry out a statistic evaluation?
Line 143 – could you provide a reference for this statement?
Table 3 – the results are of quite difficult reading.
Section 3.6 - The procedure of tests carried out after material regeneration should be described. Not completely clear what the water contained in a container was.
Table 4 – results from lines Av-5 and Av-2 seem quite odd.
Author Response
Reviewer 1.
The paper could surely be of interest for the journal readers, but in my opinion it suffers from two main drawbacks:
( i) One the one hand it is not clear the relation of this study with other studies on the same subject. Which is its innovative character? In my opinion the state of the art for what concerns the disinfection through clay complexes is not adequately described.
Answer (i)
The innovative character is emphasized in the sections “Results”, “Discussion” and “Conclusions”: In the Introduction section we have pointed out that the micelle- clay complex in its granulated form enables efficient removal by filtration of a variety of bacteria and a parasite, such as cryptosporidium from drinking water. This was documented in references 5, 8, 9 and 11. Obviously, a good filtration procedure enables to reduce the amount of harmful disinfection by products, but this requires a dedicated study. Ref. 7 demonstrates the application in purification of grey water. Ref 6 mentions removal of bacteria by a clay – polymer complex. The focus in this manuscript is on developing a new powerful and low cost regeneration procedure of the used complex, which enhances the capacity of the used complex, saves energy and has also environmental advantage due to reuse of the purification complex.
We added at the end of the Introduction section a sentence: “The regeneration, which enables a low cost reuse of the purification complex has an environmental advantage.”
(ii) On the other hand, methods are not adequately described. The results contained in most of the tables are of very difficult reading and interpretation.
Answer
Most of the Methods described in Section 2 were considered as adequate in previous articles in prestigious scientific Journals. Our response to specific comments of Reviewer 1 regarding Methods will demonstrate that in fact the methods used, which were augmented in the revised version by brief explanations are adequately described.
Punctual remarks:
Line 45 – The authors mentioned two types of granulated complexes, but they did not explain what these types are. In the same line it is not clear to what “produced” refers.
Answer
Elaboration on the two types of complexes is presented later in discussing the results of Table 2 (Line 140), whose expanded text is denoted in red:
The results in Table 2 demonstrate that MW heating removed water more efficiently from ODTMA-clay granules than from BDMHDA-clay granules, “whose critical micelle concentrations are 0.3 and 0.6 mM, respectively.”
The word “produced” was replaced by “synthesized” albeit this was done on a large scale already, and we have about 2000 kgs by now. The sentence on Line 45 was expanded:“ The micelle-clay complexes synthesized by interacting micelles of an organic cation with a small critical micelle concentration with a clay which has a relatively large surface area; they include ….of the clay…”
For which reason the regeneration by using MW could be preferred to other, more traditional, methods? Not clear if it is a completely new method or if it was tested by other authors.
Answer
Only a few articles described regeneration procedures of clay-composites. The motivation is stated in Lines 56-57: “In removing pollutants from drinking water or wastewater, it is important to reduce the processing time and costs. Regeneration of the used filter after removal of bacteria was accomplished by passing dilute solutions of NaOCl or HCl, [5-8], or by heating in an oven for 2 h at 105 0C [7-8].” This article if accepted for publication will be the first to report on the use of MW in regeneration of used complex after removal of bacteria from water. As it turned out (Results, Discussion and Conclusions sections) the MW heating procedure is the best from the point of view of removal of bacteria after regeneration , saving of time and energy and low cost of the procedure.
Line 76 – Please refer to Supplementary Materials for the description of the preparation of the granules.
Answer
We added on line 76: “see Supplementary Material.”
Line 128 – Where are shown the results described from line 128 to line 132?
Answer
The results are briefly presented without a Table, since a more extensive description focused on a conventional and MW ovens is presented in Lines 133 to 139 (including Table 1).
Table 1 – not clear the sense of the results reported in the Table. Which is the meaning of the figures reported in the three columns below “granules”?
Answer
In response we added a footnote “b” to this Table: “bThe values in the Table in columns 2-4 and 6-8 in the line starting with 0.0 are initial masses (g) of the granules before heating.“
Table 2 – a diagram should probably be a better representation of these data. In my opinion the difference between the two types of granules are not so evident. Did you perform out some replicates? Did you carry out a statistic evaluation?
Answer
We refer in Table 2 to the footnotes in Table 1, which also state the standard error. The measurements were mostly in triplicates.
Line 143 – could you provide a reference for this statement?
Answer
We were puzzled by this comment, since we provided 2 detailed references in this sentence, (8 and 17) which ends on Line 144.
Table 3 – the results are of quite difficult reading.
Answer
Several readers who are far from being experts found this table clear. (Incidentally, Reviewer 2 did not seem to have difficulties with this Table). We hope that after clarifying Tables 1 and 2, Reviewer 1 will not find this Table difficult to read. Otherwise, we are ready to add more details in response to specific questions.
Section 3.6 – (i)The procedure of tests carried out after material regeneration should be described. (ii)Not completely clear what the water contained in a container was.
Answer
- The test was by removal of bacteria by filtration.
(ii) Since 6 (or even 8) columns had to be fed by one tap, a container collected the water from the tap and pipes collected the water via a peristaltic pump. Furthermore, the container, which is open to overflow, is used as a buffer to avoid pressure fluctuations in the main water supply. The improved results in the current experiments are explained by prevention of standing water in the container and disinfection of the container and pipes before the beginning of the experiments.
Table 4 – results from lines Av-5 and Av-2 seem quite odd.
Answer
We thank the reviewer for this comment. The new Table 4 together with the footnote are presented below.
Table 4. Removal of total bacteria count by fresh micelle-clay granules: number of emerging bacteria (CFU per mL) after filtration of given volumes.α
|
Column ID |
37.5 (L) |
64 (L) |
78 (L) |
|
TAP |
1200a |
1400a |
40b |
|
Container |
1100a |
3400a |
70b |
|
M1 |
|
9 |
12 |
|
M2 |
|
1 |
28 |
|
M3 |
|
835a |
925a |
|
M4 |
|
<1 |
1 |
|
H1 |
|
5 |
5 |
|
H2 |
|
1 |
363a |
|
H3 |
|
- |
56b |
|
H4 |
|
2700* |
23000* |
|
Av of H samples |
1 |
902a |
5856* |
|
Av of M samples |
350a |
211a |
242a |
αExperiment was in September, 2019. TAP and Container refer to continuous tap and container water inlets, respectively. On the first sampling (37.5 L) all samples denoted by H were mixed together and only the mixture was sampled; similarly the same was done for samples denoted by M. The average values were also calculated for the samples taken after 64 and 78 L. The average flow rate was 6 ml/min for each column. The estimated relative errors in the number of bacteria in cases denoted by *, a, b are 50% or above, 30% and 20%, respectively; in other cases less than 10%.
English.
Reviewer 1 requested extensive improvement of English. Indeed, none of the authors of this manuscript has originated from a country whose spoken language is English, but I do not recall a request for improving the English during the last (at least) decade from reviewers of articles or from publishers of a book which appeared in 2019, coauthored by Shlomo Nir and Uri Shuali.
Reviewer 2 Report
Comment 1
The word font needs to be confirmed. (Line 98-101, Table 1 and Figure 2).
Comment 2
The authors mentioned that granulated micelle-clay complex regeneration could be accomplished by passing dilute solutions of NaOCl or HCl, or by heating in an oven for 2 h at 105 0C. However, the authors use diluted HCl and followed by heating methods. Is it important to use heating to regenerate even using diluted HCl regeneration method?
Comment 3
According to the heating methods the authors used, they compared MW heating drying with conventional oven heating drying. Microwave heating drying method could be with the function of non-surface heating by water molecules vibration heating and could penetrate into inner parts of granulated micelle-clay complex. Traditional heating method only dried mostly the surface of granulated micelle-clay complex. Did the authors use infrared heating method that is also with the function of heating inner parts of granulated micelle-clay complex?
Comment 4
For the inner parts of granulated micelle-clay complex regeneration, is heating to dry out the water is important for the second time use of filtration?
Author Response
Reviewer 2
Comment 1
The word font needs to be confirmed. (Line 98-101, Table 1 and Figure 2).
Answer
Changes in fonts were introduced by the Submission system of mdpi.
Comment 2
The authors mentioned that granulated micelle-clay complex regeneration could be accomplished by passing dilute solutions of NaOCl or HCl, or by heating in an oven for 2 h at 105 0C. However, the authors use diluted HCl and followed by heating methods. Is it important to use heating to regenerate even using diluted HCl regeneration method?
Answer
We and at least also Undebeytia in ref (6) used successfully regeneration by passing dilute solutions of HCl (we also tried NaOCl, but HCl is more economical). The study in this article demonstrates significant advantage in using MW, from the point of view of the outcome, and cost. In the second regeneration this advantage was very striking.
Let me add that when we started our tests we could not be sure of this outcome, in particular in view of occasional failures until we reached a suitable procedure.
Comment 3
According to the heating methods the authors used, they compared MW heating drying with conventional oven heating drying. Microwave heating drying method could be with the function of non-surface heating by water molecules vibration heating and could penetrate into inner parts of granulated micelle-clay complex. Traditional heating method only dried mostly the surface of granulated micelle-clay complex. Did the authors use infrared heating method that is also with the function of heating inner parts of granulated micelle-clay complex?
Answer
The traditional oven also utilized infrared heating, but we have not tested an oven which operated exclusively by infrared heating.
Comment 4
For the inner parts of granulated micelle-clay complex regeneration, is heating to dry out the water is important for the second time use of filtration?
Answer
By drying out the water, inactivation or killing of adsorbed bacteria, (which may be retained in small water pores) is secured.
Round 2
Reviewer 1 Report
I’m not sure that the authors did all the requested changes. The first drawback to which I pointed out in my first review report remains, that is the relation of this study with other studies on the same subject is not completely clear. In the same way, also the innovative character of the study was not well underlined.
Furthermore, the paper remains quite confused and of difficult reading, because both methods and results are not adequately described.
Sentence from line 45 to 47 remains unclear.
My previous remark was not addressed: Table 2 – a diagram should probably be a better representation of these data. In my opinion the difference between the two types of granules are not so evident. Did you perform out some replicates? Did you carry out a statistic evaluation?
Punctual remarks
Line 121: not clear what “other durations” means.
It is not clear to me the sense of the discussion reported in line 124. In this line the mass loss due to a conventional heating is cited but, in the previous lines, I did not find results concerning a mass loss due to a conventional heating, the authors report only the mass loss due to MW irradiation.
Author Response
The authors appreciate the expertise and dedication of the reviewers of the manuscript. Rev. 2 did not have more unanswered question after the first revision,
Second round of comments by Reviewer 1. Second revision
(i)I’m not sure that the authors did all the requested changes. The first drawback to which I pointed out in my first review report remains, that is the relation of this study with other studies on the same subject is not completely clear. In the same way, also the innovative character of the study was not well underlined.
Answer
Our impression is that Reviewer 1 is dedicated to improve and polish our manuscript, but the statements in his comments above are of general nature, and apparently led us to respond and make corrections which were not in line with his point of view, despite our long answer in round 1, and a few clarifying statements, in the manuscript, which were never required in our previous articles in prestigious scientific Journals in the areas of clay science and water purification. In order to move forward in the revision process SPECIFIC COMMENTS are needed. In an attempt to guess, I point out that the last paragraph in the Introduction (in the current version) did include a sentence and a reference (10) to a review about microbial inactivation by MW. In line with the question of Reviewer 1 in the first round we added a sentence on line 67: “This is the first study on application of MW radiation to the regeneration and reuse of clay-composites in removal of bacteria from water by filtration. As will be shown, the MW procedure turned out to be optimal in terms of removal of bacteria and saving on energy and time.”
(ii)Furthermore, the paper remains quite confused and of difficult reading, because both methods and results are not adequately described.
Answer
Again, SPECIFIC COMMENTS are needed both for elucidating passages which can benefit from easier reading and particular deficient passages in Methods and
Results.
As a guess we have expanded the description un section 2.4 (L 26)
“ Water was taken continuously from a tap into a container, which was overflown to avoid reproduction of bacteria. The filters were fed from the container by a peristaltic pump. The filters used (1.6 cm diameter x 20 cm length) were filled exclusively with 32 g of a granulated complex. Non-woven polypropylene geotextile filters (Markham Culverts Ltd., Papua, New Guinea) were placed on both ends of the column. The flow rate was 6 mL/min, i.e., a flow velocity of 1.8 m/h. Prior to each experimental run, water was added to the columns at a slow rate in an upward direction in order to eliminate air pockets and channeling. Each experiment was conducted in triplicate at least. “
In addition, we expanded subsections 2.4 Filtration.. and 2.5 Regeneration:
2.4. Filtration Experiments of TBC
Water was taken continuously from a tap into a container, which was overflown to avoid reproduction of bacteria. The filters were fed from the container by a peristaltic pump.The filters used (1.6 cm diameter x 20 cm length) were filled exclusively with 32 g of a granulated complex. Non-woven polypropylene geotextile filters (Markham Culverts Ltd., Papua, New Guinea) were placed on both ends of the column. The flow rate was 6 mL/min, i.e., a flow velocity of 1.8 m/h. Prior to each experimental run, water was added to the columns at a slow rate in an upward direction in order to eliminate air pockets and channeling. Each experiment was conducted in triplicate at least.
2.5. Regeneration
Regeneration of filters (1.6 cm x 20 cm) containing 32 g of micelle-clay complexes after filtration of TBC was carried out by passing 250 mL of a solution of 0.05 M HCl followed by washing with 500 mL of tap water, or/and by heating. Heating of filter was applied in an oven or a microwave oven; mainly two procedures were applied. (i) Heating in an oven at 120°C for 2 hours, or (ii) Heating in a MW oven for variable number of minutes , or less, depending on residual humidity, as described in Section 3. For the columns used in the current study, the wet granules from one column were removed into a small MW compatible bowl and weighed. The bowl was introduced into the MW oven for 30 to 60 s operation at 700 W. Then the granules were mixed and weighed again. In the second round the period of MW operation was reduced, e.g., to 30 s. Each consecutive round involved shorter periods of heating, until the original weight of 32 g was approached. Same procedure was applied to all the columns. It was also possible to use in one round granules from more columns, and correspondingly enhance (not linearly) heating times.
Sentence from line 45 to 47 remains unclear.
Answer
- This sentence was modified in Revision 1, but we had to leave it for a day, rather than submit the revision right away. Another reading revealed that this sentence had to be improved by splitting it into two sentences:” The micelle-clay complexes were synthesized by interacting micelles of an organic cation with a clay, which has a relatively large surface area. The two organic cations chosen were ODTMA, and Benzyldimethylhexadecylammonium (BDMHDA), whose critical micelle concentrations were relatively small.”
My previous remark was not addressed: (a) Table 2 – a diagram should probably be a better representation of these data. (b) In my opinion the difference between the two types of granules are not so evident. (c) Did you perform out some replicates? Did you carry out a statistic evaluation?
Answer
Let me show again my previous answer, and expand on it. “We refer in Table 2 to the footnotes in Table 1, which also state the standard error. The measurements were mostly in triplicates.: To facilitate reading I will present the footnotes from Table 1 again :
“a The ordinary oven was operated at 800 W, whereas the MW was operated at 700 W. The standard error of each mass was less than 1 g. bThe values in the Table in columns 2-4 and 6-8 in the line starting with 0.0 are initial masses (g) of the granules before heating.”
(b) In my opinion the difference between the two types of granules is evident. Here are a few examples. The results in Table 2 indicate that after 10 min the masses (g) are 50.7 and 52.7 for ODTMA and BDMHDA heated granules; after 12 min the corresponding values are 45 and 47.1 g, and after 16 min the corresponding values are 40.2 and 42.9 g. (c) The measurements were mostly done in triplicates as I stated last time, and in certain cases in duplicates. The differences are more than twice the standard error, which was less than 1 g.
These differences were discussed in the paragraph before Table 2 and are in line with previous published results on cation release from the two complexes.
- We preferred to introduce this topic in a set of associated Tables, in order to exhibit clearly the numbers. The manuscript includes quite a few Figures. Hence, we cannot be blamed for using Tables exclusively.
Punctual remarks
Line 121: not clear what “other durations” means.
Answer
I copy 3 lines; in the third line a small correction was introduced:
A sample of 5 grams of ODTMA- or BDMHDA-clay granules was subjected to the MW irradiation for 90 s, 180 s, 270 s, 360 s and 450 s. After each duration, sample mass was measured. Then immediately the sample was put in MW oven for another duration.
It is not clear to me the sense of the discussion reported in line 124. In this line the mass loss due to a conventional heating is cited but, in the previous lines, I did not find results concerning a mass loss due to a conventional heating, the authors report only the mass loss due to MW irradiation.
Answer
The first paragraph in section 3.1 includes a reference to 3 previous studies, which is useful for the readers. In Table 1., which is placed just a few lines later, we present an extensive comparison between MW and ordinary oven heating of ODTMA-clay granules.
English.
We approached a scientist (Ph,D in Theoretical Physics), Dr. Yehuda Alexander for reading the manuscript and suggesting improvement of the English.
Corrections following his comments were introduced into the revised manuscript in blue.
Reviewer 2 Report
No other comments
Author Response
Reviwer 2
No other comments.
Answer:
The authors appreciate the expertise and dedication of the reviewers of the manuscript. Rev. 2 did not have more unanswered question after the first revision.